# Energetic Bio-Activation of Some Organic Molecules and Their Antioxidant Activity in the Pulp of the Moroccan Argan Tree *«Argania spinosa* L.*»*

**DOI:** 10.3390/molecules27103329

**Published:** 2022-05-22

**Authors:** Ayoub Mourjane, Hafida Hanine, El Mustapha El Adnany, Mourad Ouhammou, Nadia Hidar, Bouchra Nabil, Ahcène Boumendjel, Khalid Bitar, Mostafa Mahrouz

**Affiliations:** 1Laboratory of Bioprocesses and Bio Interfaces, FST Beni Mellal, University Sultan Moulay Slimane, Beni Mella 23000, Morocco; ayoubmourjane@gmail.com (A.M.); h.hanine@usms.ma (H.H.); 2Laboratory of Material Sciences and Process Optimization, Faculty of Sciences Semallaia, Cadi Ayyad University, Marrakesh 40000, Morocco; eladnanyelmustapha@gmail.com (E.M.E.A.); ouhamoumourad@hotmail.com (M.O.); nadia.hidar@gmail.com (N.H.); mahrouz@uca.ac.ma (M.M.); 3Faculty of Applied Sciences, University Sultan Moulay Slimane, Fkih Ben Saleh, Beni Mella 23000, Morocco; nabiilaa23@gmail.com; 4Laboratoire Radiopharmaceutiques Biocliniques (LRB), INSERM U1039, Faculté de Médecine La Tronche, Université Grenoble Alpes, 38000 Grenoble, France; 5IRCOS Laboratory, ZI Al-Massar, Marrakesh 40000, Morocco; bitar.k@ircoslaboratoires.com

**Keywords:** *Argania spinosa* L., thermal activation, antioxidant activity, polyphenols, chemical analysis

## Abstract

*Argania spinosa* L. Skeels is an emblematic tree in Morocco, known worldwide for its medicinal and nutritional value. Its fruits contain kernels used to prepare an edible oil, the leaves are used to feed livestock, and its wood is used as fuel. If the oil acquires high importance, the other components of the fruit of the argan are undervalued. Our objective is to invest the waste of the argan industry. Particularly, our study aimed to assess the effect of thermal activation of argan pulp on its therapeutic value, its phenolic profile and its functional and physicochemical properties. After heat treatment, the HPLC analysis for the average total phenolic content varied from 2% to 37%, depending on temperature. The antioxidant activity was increased with heat treatment. Higher values of antioxidant activity, polyphenol and pigment content were recorded at 70 °C. Functional properties analysis indicated that water solubility index and water absorption capacity were significantly affected by heat stress. Physicochemical analysis showed that moisture content, titratable acidity and soluble solids were affected.

## 1. Introduction

The argan tree (*Argania spinosa* L. *Skeels*) is a monotypic tree, growing endemically in the southwestern Moroccan area, with extraordinary ethnobotanical value [1]. This tree is mainly exploited for the production of its edible fruit oil, which has multiple medical virtues, including hepatoprotective, hypocholesterolemic and atherosclerotic effects [2].

On the industrial level, the continuous non-destructive athermal extraction of argan oil of good quality leads, unfortunately, to the generation of large amounts of waste. The pulp, the cake and the hull represent 45, 35 and 16% of the waste, respectively. These wastes are used as fuel and livestock feed. The pulp of the fruit of the argan tree is a material rich in minerals, fiber, polysaccharides, bioactive (Omega 3, 6, and 9) and secondary metabolites, including flavonoids [3] and saponins [4].

Heat treatment can affect fruit quality parameters, such as coloring, vitamins, texture and sensory properties [5]. Moreover, the heat treatment may alter the integrity of bioactive compounds [6]. In this regard, the degradation of phenolic compounds in fruits and vegetables depends on the type of food, time and temperature of treatment [7]. Thus, the evaluation of the therapeutic and physicochemical quality of fruits after drying is important.

Nowadays, there are well-established references on the drying impact on various vegetables and fruits [8,9], including Moroccan sweet cherry [10], rice [11], green bell pepper, green bean and squash [12], potatoes [13], pea, bean and artichoke byproducts [14] and cactus [15]. To our knowledge, there are no published scientific data on the effect of heat treatment on the chemical composition and anti-free radical capacity of the pulp of argan. Herewith, in this study, we aimed to investigate the effect of temperature on the functional, physicochemical and antioxidant activity of the argan pulp powder and identify the appropriate temperature for food and pharmaceutical industries added value. The principal component analysis (PCA) was conducted to study the variability and correlations between the parameters studied, and to discriminate the different temperatures according to their antioxidant activities and other parameters. PCA is a new approach to study bio-activation energy, by conventional temperature, solar or gamma radiation. The input of the PCA approach was confirmed for the gamma radiation effect during the conservation of the clementine and leaves of cactus rackets, where the contents of sugar and polyphenols were increased [16,17].

Our study was focused on the impact of heat treatment on the pulp of the Moroccan argan tree «*Argania spinosa* L.» on the chemical composition, antioxidant activity and physicochemical properties. The ultimate goal was to provide valuable data to valorize the product in terms of sanitary, nutritional and bio-therapeutic quality.

## 2. Results and Discussion

### 2.1. Physicochemical and Functional Analysis

The variation in physicochemical and functional properties of the powder as a function of the processing temperature is presented in Table 1. The study of moisture tenure in food is essential for its preservation and stability, needed for commercial and legal reasons [18]. In addition, moisture tenure is a property that affects the physical, chemical and microbiological quality of foods. The control was out of the limit of storage conditions in terms of moisture content 14.30 ± 0.25%. The heat treatment showed a significant impact, with a statistically significant relationship (*p* < 0.05), on the moisture percentage, with a range of 6.73 to 3.14%, which means the powder is within the standards of storage at 60 °C. The impact of temperature on ash content is not statistically significant (*p* > 0.05); this may be due to the resistance of mineral elements to heat stress. However, heat treatment does not have a significant effect on pH and titratable acidity, while the effect on insoluble solids is remarkable. Thermal treatment softens plant tissues and leads to the inactivation of endogenous pectinolytic enzymes. At low temperatures, the enzymes contribute to the fragmentation of pectin chains, through the action of the enzymes pectin methylesterase (PME), and the enzyme polygalacturonase (PG) acts on the segments of the pectin chain undergoing demethylation by PME. However, increasing temperature causes inactivation of both PME and PG enzymes [19]. Regarding the effect on pH, Simonen et al. 2007 showed that the variation in acidity depends on the variation of nitrogen in the plant matrix, so a stable amount of nitrogen causes a stability of acidity [20].

The functional properties versus processing temperature are presented in Table 1. Water-holding capacity is an indispensable functional property in the development, cohesion and maintenance of the viscosity of the pharmaceutical and food product. The water absorption capacity (WAC) was significantly changed according to the treatment temperature (*p* < 0.05). It was reported that the variation in WAC of the powder is attributed to the number of hydration positions, pH, and the presence of lipids and carbohydrates that allow more interactions with water through hydrogen bonding [18]. Pearson’s test showed that WAC is negatively correlated with minimum gelation concentration (LGC) and swelling capacity (SC) (r = −0.05, r = −0.40), respectively. This may be due to the presence of polysaccharide chains that associate perfectly with water. The swelling and water absorption capacities also depend on the composition and structure of the powder. The impact of heat treatment on the water solubility index (WSI) is discriminative (*p* < 0.05), ranging from 54.42 to 46.39%, but still remains very high, which limits the separation of particles adequately after centrifugation and leaching of water-soluble components [21].

The high fat retention capacity makes the powders suitable for flavor and mouthfeel improvement when used in food preparation. No significant variation (*p* > 0.05) was noticed for fat absorption for all processing temperatures, which may be due to the variation in the interaction potential with water and oil and conformational characteristics. Thus, the correlation matrix showed that the fat-retention capacity is negatively correlated with LGC, SC and WAC. Overall, temperature is an important parameter that could create a significant change in powder compositions, fibers, soluble proteins, carbohydrates, lipids and other substances.

### 2.2. Determination of Color and Chlorophylls

The data obtained from the color measurement are presented in Figure 1. It showed that the influence of the various conditions of treatments on the color of pulp powder is significant for the three parameters a*, b* and L*. From 25 to 60 °C, it was observed that the thermal stress increased the clarity (L*) of the powder, with a maximum estimate of 58.91 ± 0.26 at 60 °C.

Starting from 70 °C, we noticed a browning of the vegetable matter. In addition, the powder has a tendency towards a red color, as indicated by the positive variation in the a* value, from +27.49 to +50.43, while the b* value remained more or less similar, whatever the processing conditions, showing a yellow appearance. The latter can be used as a natural coloring additive in a food or cosmetic product. Statistical analysis showed a significant positive correlation between temperature and a* and b* values (r = 0.60 and r = 0.70, respectively), while the correlation with the brightness L* was weakly negative (r = −0.04). The color change could be due to the destruction of chlorophyll to pheophytin, which often occurred with green vegetables subjected to heat treatment [22] and the non-enzymatic browning, known as products of Maillard reaction that occurs during drying and is favored at high temperatures [23].

The effect of temperature on chlorophyll a and b content was studied and the results are shown in Figure 2. At 70 °C, higher contents of chlorophylls a and b were observed: 1.52 ± 0.04 and 2.53 ± 0.06 mg/g DM, respectively. Chlorophylls are the most common photosynthetic pigments in the plant kingdom, which gives an indirect estimate for the nutrient profile due to the high incorporation of nitrogen by this pigment. This relationship is used to determine the photosynthetic potential [24]. It was reported that chlorophylls and pheophytins had high anti-free radical activities against DPPH and a bleaching effect of beta-carotene, due to the presence of magnesium, which affects the electron donor capacity in the porphyrin system.

### 2.3. Phenolic Analysis 

#### 2.3.1. Total Polyphenol Content, Total Flavonoid Content

The results for the impact of heat treatment on total phenolic compounds (TPC) and total flavonoids compounds (TFC) of argan pulp powder are presented in Figure 3. The TPC of the pulp dried at 25 °C showed an average of 18.83 ± 0.56 mg GAE/1 g dry matter. However, the pulp treated at 70 °C showed a significant increase (a maximum average of 379.03 ± 0.97 mg GAE/1 g of DM. This increase was followed by a decrease to 262.18 ± 0.96 mg GAE/1 g DM at 100 °C. The treated pulp was significantly affected by temperature, as we observed an evolutionary peak up to 379.03 ± 0.97 mg GAE/1 g DM at 70 °C, followed by a decrease to 262.18 ± 0.96 mg GAE/1 g DM at 100 °C. This variation can be explained by the optimal stability of the polyphenoloxidase (PPO) enzyme, and the phenol hydroxylation mechanism [25]. The PPO activity remains high at 70 °C and pH = 3.9, which causes an important accumulation of CPTs, while from 80 °C and higher, a loss of PPO activity is observed, which is in agreement with the results reported for cherries [10,26]. On the contrary, the flavonoid content as a function of temperature showed a significant increase, with a maximum estimate at 100 °C 28.80 ± 1.27 mg CE/1 g DM.

#### 2.3.2. Molecular Analysis by HPLC

HPLC was used to monitor the impact of temperature on phenolic compounds. The results are presented in Table 2. Four phenolic acids (protocatechuic acid, ferulic acid, chlorogenic acid, sinapic acid), one phenolic aldehyde (vanillin), three flavonoid aglycones (catechin, quercetin and kaempferol) and one flavonoid glycoside (rutin) were detected. The results showed that the heat treatment has a significant effect on the chemical composition, qualitatively and quantitatively. The total concentration of phenolic compounds varied from 257.6 to 1963.7 mg/100 g DM. The evolution of the concentration of protocatechuic acid is proportional to the treatment temperature (from 4.4 to 595.8 mg/100 g DM, respectively), while other phenolic compounds are present in an opposite trend when the temperature applied is 40 °C (Figure 4). The concentration of rutin, catechin, quercetin, epicathechin and chlorogenic acid is maximum (575.4 ± 1.36; 250.1 ± 1.23; 46.9 ± 0.27; 651.65 ± 1.21 and 345.2 ± 1.07 mg/100 g DM respectively) at 40 °C and then decreases with temperature. However, the concentration of kaempferol is maximal at 70 °C. These results may be due to the release of quercetin from the quercetin-3-O-rutinoside (rutin) and degradation to protocatechuic acid [27]. It was shown that quercetin-3-O-rutinoside degrades to quercetin at 70 °C. The latter, under the effect of temperature in the presence of oxygen, is easily oxidized to phloroglucinol carboxylic acid and other phenolic compounds. It was observed that epicathechin and catechin can undergo rapid degradation at 70 °C and complete degradation occurs at 100 °C. Depending on their structure, flavonoids are more or less sensitive to heat treatment. Glycosylated flavonoids are more resistant to heat treatment than aglycone flavonoids [27]. Degradation depends on strength and structure, which explains the resistance of rutin compared to epicathechin and catechin. However, vanillin is detected only in the control sample (25 °C) and high temperatures with very low concentrations (5.9; 3.1; 3.52 and 2.04 mg/100 g DM), respectively. The identification of vanillin at 80, 90 and 100 °C may be due to the thermal degradation of ferulic acid. The small quantification of ferulic acid in all samples and its low concentration at 90 and 100 °C compared to other temperatures may be due to its conversion to vanillin or its demethylation to caffeic acid [28,29,30]. It should be highlighted that our samples are very rich in flavonoids; however, the majority of phenolic acids, such as caffeic acid, salicylic acid, vanillic acid and p-coumaric acid, were not identified in such samples. This observation is in full agreement with previously reported results [31]. (All HPLC figures in Appendix A).

### 2.4. Antioxidant Activity

The antioxidant activity at each temperature was measured by DPPH, FRAP and ABTS free radical scavenging tests (Figure 5) [32,33]. The variation in free radical scavenging activity was significant for all three tests. The maximum activity estimated by the FRAP (95%) method was recorded for the temperature 70 °C. For the ABTS method, a high AAO was observed: 95.48 ± 4.47%; 97.01 ± 2.28% and 98.28 ± 1.89% at 60, 70 and 80 °C, respectively. The DPPH test showed a clear correlation between the sheathing of free radical scavenging activity and temperature. It is evident that the strong activity of the extracts in capturing free radicals is attributed to their richness in phenolic compounds, which have the highest content of molecules, including polyphenols, flavonoids and tannins [31]. This is explained by the high correlation between CPT and DPPH, FRAP and ABTS (r = 0.74, r = 0.73 and r = 0.27, respectively). The correlation between the anti-free radical activity obtained from the FRAP and DPPH assay is strongly positive (r = 0.69), given that the correlation with the ABTS assay is weakly negative (r = −0.08). However, the correlation between DPPH and ABTS is very weak (r = 0.05). This proves that phenolic compounds are mainly responsible for the antioxidant activity of the argan pulp powder, especially for the temperature 70 °C, with the tests ABTS and FRAP.

### 2.5. GC-MS Analysis

The oils extracted from the treated argan pulp were analyzed, as an indicator of quality, before and after the application of heat stress, and their compositions were studied. Spectral analysis using GC-MS of eight samples identified new molecular peaks at 40 °C and higher (Table 3). A slight difference in the oil compositions was noticed, compared to the control for the samples treated at 40 and 50 °C. However, the compositions of the oils in the samples exposed to 60, 70, 80, 90 and 100 °C were significantly changed. Indeed, the identification of sterol nephews, alpha-amyrin and olean-12-en-3-ol acetate, were mainly present. Other unidentified components were detected in significant amounts, corresponding to retention times of 56.91 and 57.70 min. The percentages of alpha-amyrin and olean-12-en-3-ol acetate are more remarkable at medium-high temperatures (70 and 80 °C). It was reported that the mixture between alpha-amyrin and olean-12-en-3-ol acetate has a potential antihyperglycemic and hypolipidemic effect [34]. These results could be exploited to prove the added value of heat treatment for argan pulp. 

### 2.6. Principal Component Analysis (PCA)

The PCA was used to study the thermal variability and trend as a function of functional, physicochemical and biochemical variables. The impact of thermal stress on functional properties, physicochemical properties, bioactive content and anti-free radical activity were studied at different temperatures (Figure 6). The analysis showed a strong positive correlation between chlorophylls a, chlorophylls b, anti-free radical activity tests (DPPH and FRAP), color measurements (b*) and swelling capacity (presented on the right of the diagram). Moreover, strong negative correlation was observed between titratable acidity, water solubility index (WSI), pH, water holding capacity, WAC, moisture and ABTS test (presented on the left side of the diagram). The study of correlation for temperature with the different parameters revealed the presence of variability between all the treatment temperatures (Figure 6). From the distribution of the treatment temperature and the studied parameters, we noticed that the temperature had a strong correlation with total polyphenols, total flavonoids, chlorophyll a, DPPH test and LGC property. However, it had negative correlation with pH and humidity. Temperatures of 40 and 50 °C have a correlation with DPPH, humidity, titratable acidity and WSI. Consequently, the temperatures (60, 70 and 80 °C) applied to the powder were correlated with the tests (DPPH and FRAP), the total polyphenols, the deviation in the flow of the powder, the chlorophylls and the swelling property (SC). However, the highest temperatures (90 and 100 °C) were correlated only with total flavonoids. It should be noted that the medium-high temperatures (70, 80 °C) had a strong association with the bioactive and anti-free radical activity of the argan pulp powder, which proves the added value of heat treatment in the pharmaceutical and food industries. 

## 3. Conclusions

Heat treatment leads to a differential distribution of the chemical composition and bioactive compounds in argan pulp powder, depending on the temperature. This study showed that the temperature had a significant effect on the functional, physicochemical and antioxidant properties of the powder. The temperature set at 70 °C offers a high antioxidant potential, high polyphenol and chlorophyll content. These results demonstrated interest in the thermal activation process for the improvement of the therapeutic properties of the pulp powder and would help the food and pharmaceutical industries to develop products with an appropriate sanitary quality and optimal functional and nutritional applications.

## 4. Materials and Methods

### 4.1. Materials and Instruments Used

In this study, pH was measured by using a pH meter (Hanna Instruments). For measuring water content, a refractometer (DR 6000, A. Krus Optronic GmbH, Hamburg, Germany) was used. UV-visible spectra were recorded by using UV-vis spectroscopy (Pg Instruments, T80). The chromatographic separation was performed with apparatus (Knauer, Berlin, Germany) equipped with a quaternary pump, auto-sampler and column oven. A Kinetex C18 reverse phase column (100 × 4.6 mm, 2.6 μm particles) and a DAD detector, programmed to an acquisition interval of 200–700 nm, were used for the proposed method. The equipment used for GC-MS analysis included a gas chromatograph coupled to a mass spectrum (Polaris Q MS with ionic trap), the CPG is equipped with an apolar capillary column (Rtx-1 (60 m × 0.22 mm); thickness of the film 0.25 µm), coupled to a Perkin Elmer Turbo Mass detector. Principal component analysis (PCA) was performed using XLSTAT software (2016, France).

#### Plant Material

The argan pulp was recovered from the argan oil production unit of IRCOS Laboratory. The fruits of *Argania spinosa* L. were harvested in the region of Essaouira, Morocco. These fruits were mechanically peeled during oil production. A voucher was deposited in our laboratory under the reference LOT CA128. Afterwards, the pulp was thermally treated in a ventilated oven, non-solar drying at different temperatures (from 40 to 100 °C) until the total dehydration at atmospheric pressure. Each sample was ground to powder with an expensive grinder (25,000 r/min) and stored in amber, well-sealed bags at 20 °C.

### 4.2. Methods

#### 4.2.1. Functional Properties

Water absorption capacity (WAC) and oil absorption capacity (OAC) were determined according to the method described by [35]. Briefly, one gram of the powder sample was independently mixed with distilled water or soybean oil (10 mL) in a centrifuge tube and allowed to stand at room temperature (25 ± 2 °C) for 1 h. The samples were centrifuged (200 rpm for 30 min). The WAC or OAC was expressed as the percentage of water or oil absorbed per 1 g of powder. The powder swelling capacity (SC) was determined using the protocol described by [36] and the results were expressed in (mL). The method of [37] was used to determine the minimum gelation concentration (LGC). The solubility index (WSI) in water was calculated according to the method of [38]. The solubility index was calculated in g per 100 g of sample powder on a dry-weight basis. The WSI was calculated by the following equation: WSI (%) = Ws/W_DS_ × 100, where Ws: weight of dissolved solids in the supernatant; W_DS_: weight of dry solids.

#### 4.2.2. Physicochemical Analysis

After each treatment, the samples were analyzed for percent moisture, total ash, total soluble solids, pH and titratable acidity. The percent moisture of each sample was determined by drying in oven at 105 ± 2 °C for up to 24 h. Ash determination was performed by incinerating the samples in a muffle furnace at 550 ± 5 °C for 3 h and calculated as the mass difference [39]. For measurement of pH and soluble solids (Brix), 1 g of each sample was added separately to 10 mL of distilled water, homogenized and filtered, the pH was measured and the refractometer was used to evaluate the moisture level. The titratable acidity was evaluated by titrating 10 mL of aliquot sample to pH 8.2 with 0.1 N NaOH solution. The results were expressed as % citric acid using the following formula % = [(N × V × mEqcitric acid)/W]. N: the concentration of NaOH; V: the volume of NaOH; W: the weight of the sample; and mEq: the milliequivalents of citric acid (0.064) [40]. All samples were analyzed in triplicate.

The determination of chlorophyll a and chlorophyll b was performed in a whole pigment acetone extract by UV-vis spectroscopy. Using the equations given below and according to known methods [41]. The content of chlorophyll a and chlorophyll b in the acetone extracts were calculated as follows: chlorophyll a = 11.24 × A662 − 2.04 × A645 g/mL; chlorophyll B = 20.13 × A645 − 4.19 × A662 g/mL. Results were reported in mg/g on a dry-weight basis.

The brightness L*, redness a* and yellowness b* are the parameters to be followed to measure the development of the color of the powder according to the temperature of treatment; they were carried out using a chromameter (Model CR200, Minolta). The color is defined in the colorimetric space by the spatial coordinates of the color (L*, a* and b*) where the value L* indicates the lightness of powder, 0–100 represents the dark to light, the value a* indicates the degree of the green-red color, if the value of a* is positive and higher the coloration tends towards the red color. The value of b* presents the degree of blue-yellow color, with a higher value of b* indicating more yellow [23].

#### 4.2.3. Total Polyphenol Content, Total Flavonoid Content and Antioxidant Activity

The previously prepared extracts underwent a biochemical analysis that studied the total phenol and flavonoid content as well as the antioxidant activity. The polyphenol content was determined using the protocol described previously [42]. Briefly, an extract of 0.2 mL diluted in 0.3 mL of distilled water was mixed with 2.5 mL of Folin’s reagent (ten-fold dilution). After 2 min, 2 mL of sodium carbonate (7.5%) was added. The mixture was incubated in the dark at 45 °C for 15 min, and after cooling, the absorbance was recorded at 760 nm against the blank. The phenolic content of each extract was calculated from the calibration curve, expressed as milligrams of gallic acid equivalent (used as standard phenol) per gram of dry matter (mg EAG/g DM). The flavonoid content of the methanolic extracts of the pulp were estimated according to reported methods [43]. A 1.5 mL aliquot of the sample was added to 1.5 mL of AlCl_3_ reagent (2%). After 30 min of incubation in the dark, the absorbance was recorded at 430 nm. A catechin calibration curve was used to calculate the concentration of flavonoids in each extract, the results are expressed as mg of catechin per one gram of dry matter. The evaluation of the effect of temperature on the any-radical power was done using three in vitro methods: DPPH, ABTS and FRAP. The method reported by Abdel-Hameed et al. [44] was used to test the effectiveness of the extracts on the free radical DPPH. Briefly, 1 mL of 0.1 M DPPH solution in pure methanol was mixed with 1 mL of extract from each sample. The reaction mixture was incubated in the dark for 30 min and optical density recording was performed at 517 nm against the blank. For the control, 1 mL of methanolic solution of DPPH was mixed with 1 mL of methanol. The percentage of DPPH inhibition was determined according to the following formula: % inhibition = (A control − A sample)/A control) × 100, where A sample: the absorbance of the sample with DPPH; A control: the absorbance of the DPPH solution with methanol.

The iron reduction assay (FRAP) was performed according to the protocol described by [43]. Briefly, 100 µL of sample was combined with 3 mL of FRAP reagent, with the latter freshly prepared by mixing 300 mM sodium acetate buffer (pH = 3.6), to 1 mM TPTZ (2,4,5-tripyridyl-S-triazine) in 40 mM HCl and FeCl_3_ in a proportion 10:1:1 (*v:v:v*). After incubation at 37 °C for 30 min, the absorbance at 593 nm was measured. The absorbance deference between the selected reading on final and blank reading was calculated for each sample. The % FRAP inhibition was calculated based on the OD of the control and sample.

For the ABTS assay, we followed the method reported by [45]. ABTS+ radical was generated by the reaction of 7 mM ABTS+ and 2.45 mM potassium persulfate, and after incubation at room temperature in the dark for 16 h, the solution was diluted to an optical density of 0.70 ± 0.02 at 734 nm. Then, 1 mL of ABTS+ solution was added to 10 µL of samples independently, and the mixture was shaken well and incubated for 30 min. The absorbance was recorded at 734 nm. The inhibition (%) of ABTS+ was calculated based on the OD of the control and sample.

#### 4.2.4. Molecular Analysis by HPLC

The identification and quantification of phenolic compounds by high-performance liquid chromatography (HPLC) was performed according to known methods [46]. Seventeen phenolic compounds were used as standards (Table 2). Initially, 100 g of the plant powder was extracted with methanol, and then the extract was evaporated. The sample (1 mg/mL) of the evaporated crude extract was used for HPLC analysis and its compounds were quantified by comparing them with the standards. 

Gradient separation was created from solvent A (0.1% aqueous formic acid solution) and solvent B (methanol) as follows: 0–3 min, linear gradient from 5 to 25% B; 3–6 min, at 25% B; 6–9 min, from 25 to 37% B; 9–13 min, at 37% B; 13–18 min, from 37 to 54% B; 18–22 min, at 54% B; 22–26 min, from 54 to 95% B; 26–29 min, at 95% B; 29–29.15 min, return to initial conditions at 5% B; and 29.15 to 36 min, at 5% B. The mobile phase flow rate was 1 mL/min. For quantitative analysis, calibration curves were prepared from different standard compounds using the areas of the mass peaks obtained from the chromatograms at a concentration between 0.5 and 1 g/L. The results are expressed as mg per 100 g DM. Under these conditions, the chromatographic separation showed a significant impact of temperature on the phenolic compositions at a time <30 min.

#### 4.2.5. GC-MS Analysis

This method was used for qualitative analysis of lipidic extracts. It was carried out with the help of the center of analysis CAC-FSSM (Faculty of Science Semlalia—Marrakech). Helium was the mobile phase with a start of (1 mL/min), exerting a pressure of 5 kPa. The temperature of the injector was 250 °C, that of the detector 280 °C. The programming of the temperature was in an interval from 70 to 250 °C at 5 °C/min, then in a 15 min step at 250 °C. The injection is performed by split mode with a division ratio of 1/50. The quantity of sample injected was 1µL. The detection was performed by a quadripolar filter analyzer. Molecules were bombarded by an electron beam (70 eV). Mass spectra were obtained in the 35–350 Da mass range by electron impact. The computer comparison of the spectrum of an unknown peak with one or more reference libraries allowed its identification. The similarity of the unknown and reference spectra was sufficient and the retention indices were identical under comparable operating conditions [47].

### 4.3. Statistical Analysis

The results of this work are presented as the mean +/− standard deviation of triple determinations and analyzed by analysis of variance (ANOVA). Tukey’s test and Pearson’s correlation were performed for the comparison of the mean at a 5% significance level and to evaluate the relationship between the parameters. Throughout the text, *p* < 0.05 means a statistically significant relationship.

## Figures and Tables

**Figure 1 molecules-27-03329-f001:**
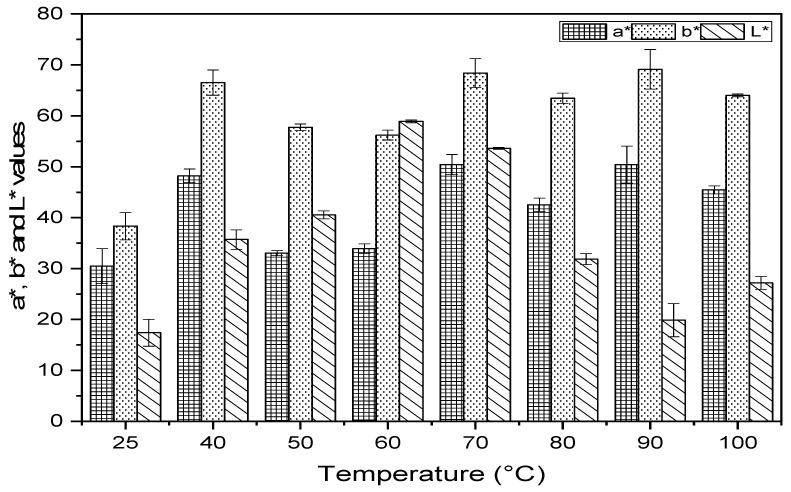
Impact of temperature on the color (values a*, b* and L*) of the powder of the argan pulp.

**Figure 2 molecules-27-03329-f002:**
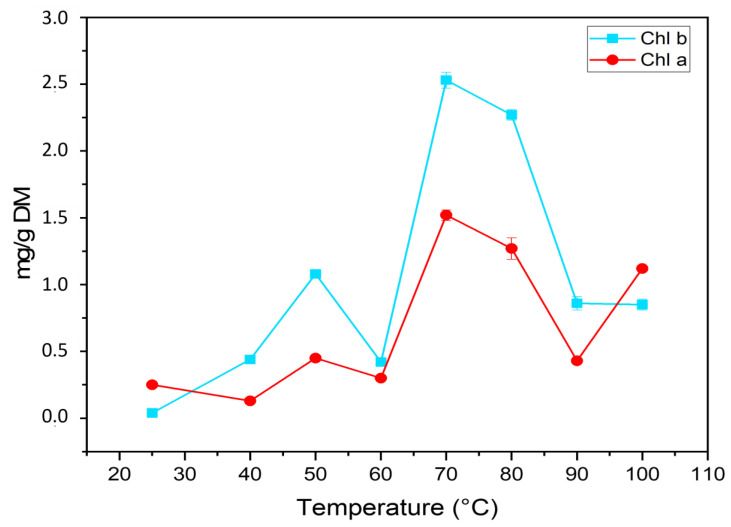
Impact of temperature on the content of chlorophylls a and b in argan pulp powder (mg/g dry weight).

**Figure 3 molecules-27-03329-f003:**
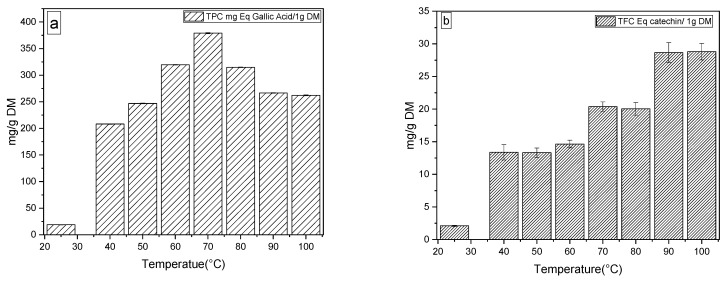
(**a**) Profile of total polyphenols (TPC) and (**b**) total flavonoids (TFC) as a function of temperature.

**Figure 4 molecules-27-03329-f004:**
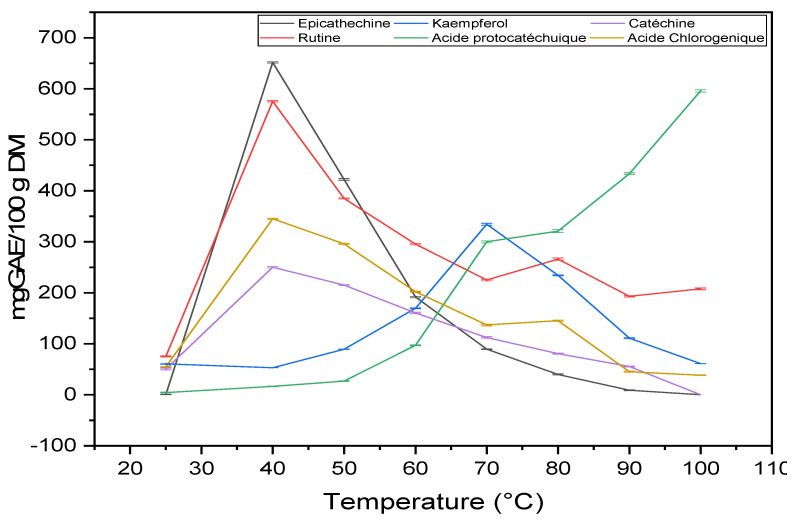
Degradation profile of phenolic compounds (mg/100 g DM) as a function of temperature.

**Figure 5 molecules-27-03329-f005:**
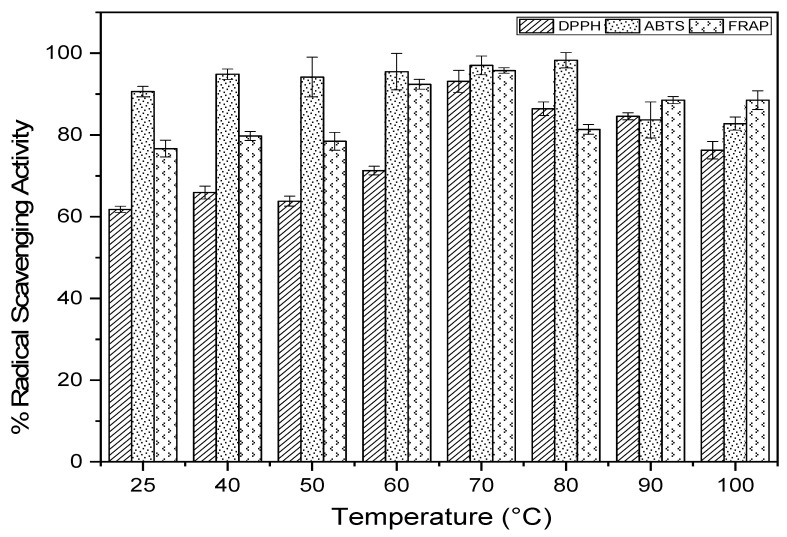
Antioxidant activity determined by DPPH, ABTS and FRAP assay from different temperatures.

**Figure 6 molecules-27-03329-f006:**
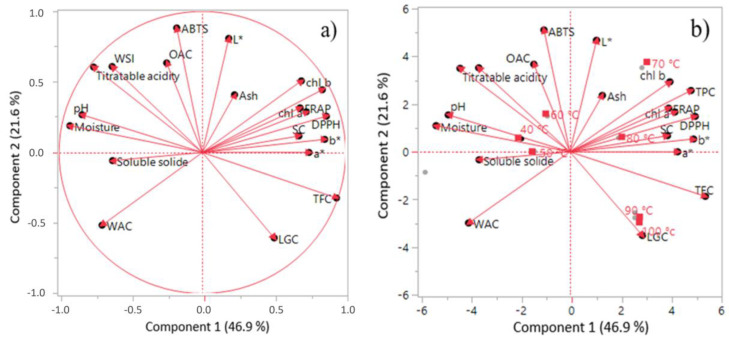
Principal Component Analysis. “(**a**) Presents the distribution of the studied parameters and the relationships between them”; “(**b**) presents the distribution of the treatment temperature with the studied parameters”.

**Table 1 molecules-27-03329-t001:** Physicochemical and functional properties of eight temperatures of treatment of argan pulp powder.

Temperature (°C)	25	40	50	60	70	80	90	100
Moisture (%)	14.3 ± 0.25	6.73 ± 0.17	5.64 ± 0.30	5.25 ± 0.31	5.00 ± 0.08	4.87 ± 0.21	3.74 ± 0.30	3.14 ± 0.26
Ash (%)	4.08 ± 0.04	4.23 ± 0.06	4.30 ± 0.21	4.15 ± 0.13	4.25 ± 0.10	4.16 ± 0.10	4.10 ± 0.09	4.20 ± 0.06
pH	4.12 ± 0.05	4.12 ± 0.10	4.11 ± 0.18	4.01 ± 0.13	4.16 ± 0.19	4.19 ± 0.09	4.19 ± 0.18	4.09 ± 0.11
Soluble solids (°Brix)	6.22 ± 0.18	6.07 ± 011	5.07 ± 0.12	3.23 ± 0.12	3.87 ± 0.26	4.17 ± 0.12	2.13 ± 0.12	5.33 ± 0.15
Titraible acidity (%citric acid)	0.34 ± 0.02	0.30 ± 0.01	0.27 ± 0.01	0.27 ± 0.01	0.29 ± 0.01	0.19 ± 0.01	0.14 ± 0.01	0.14 ± 0.01
WAC (%)	12.17 ± 0.75	11.17 ± 1.04	11.93 ± 1.01	11.43 ± 0.75	9.21 ± 1.06	9.40 ± 0.53	10.83 ± 0.76	11.41 ± 0.52
OAC (%)	12.37 ± 1.59	11.4 ± 0.46	12.1 ± 0.17	13.2 ± 0.45	12.6 ± 0.46	11.9 ± 0.07	11.86 ± 0.57	10.97 ± 0.50
LGC (%)	16.00 ± 0.00	16.00 ± 0.00	18.00 ± 0.00	16.00 ± 0.00	16.00 ± 0.00	18.00 ± 0.00	18.00 ± 0.00	18.00 ± 0.00
WSI (%)	54.42 ± 0.49	55.33 ± 0.44	50.55 ± 0.39	51.36 ± 0.47	52.63 ± 0.52	52.12 ± 0.24	47.98 ± 0.40	46.39 ± 0.37

Values are means ± standard deviation (*n* = 3).

**Table 2 molecules-27-03329-t002:** Content of individual polyphenolic compounds of eight samples of treated argan powder determined by HPLC-DAD (mg/100 g DM).

Temperature (°C)	RT (min)	25	40	50	60	70	80	90	100
Protocatechuic acid	6.48	4.4 ± 0.9	16.6 ± 0.6	26.8 ± 0.7	96.9 ± 0.9	299.9 ± 2.1	320. ± 2.85	433.6 ± 1.6	595.8 ± 2.1
Caffeic acide	14.19	ND	ND	ND	ND	ND	ND	ND	ND
Ferulic acid	19.11	5.3 ± 0.4	24.6 ± 0.1	28.16± 0.2	33.6 ± 0.1	ND	ND	6.3 ± 0.2	5.3 ± 0.8
Hesperidin	20.64	ND	ND	ND	ND	ND	ND	ND	ND
Salicylic acid	21.76	ND	ND	ND	ND	ND	ND	ND	ND
Vanillic acid	13.91	ND	ND	ND	ND	ND	ND	ND	ND
Catechin	10.97	50.5 ± 1.6	250.1 ± 1.2	215.0 ± 1.0	160.5 ± 1.5	112.5 ± 1.6	80.5 ± 0.1	55.3 ± 0.9	ND
Chorogenic acid	12.07	54.9 ± 0.9	345.3 ± 1.1	296.0 ± 1.	202.1 ± 0.9	136.8 ± 1.9	145.3 ± 1.1	45.3 ± 1.1	38.4 ± 0.6
Epicathechin	13.65	0.45 ± 0.3	651.6 ± 1.2	422.1 ± 1.6	191.6 ± 1.5	89.5 ± 0.6	39.8 ± 0.1	8.9 ± 0.8	ND
Vanillin	15.33	5.98 ± 0.5	ND	ND	ND	ND	3.1 ± 0.2	3.5 ± 0.2	2.0 ± 0.1
*p*-Coumaric acid	18.54	ND	ND	ND	ND	ND	ND	ND	ND
Sinapic acid	19.12	ND	ND	ND	22.8 ± 0.6	33.7 ± 0.4	13.9 ± 0.6	ND	ND
Naringin	20.49	ND	ND	ND	ND	ND	ND	ND	ND
Rutin	21.69	75.3 ± 1.02	575.5 ± 1.3	385.2 ± 1.0	295.5 ± 1.4	225.2 ± 1.5	266.1 ± 2.1	193.1 ± 1.4	207.7 ± 1.8
Quercetin	26.61	ND	47.0 ± 0.3	36.8 ± 0.3	29.0 ± 0.6	16.4 ± 0.4	6.5 ± 0.9	2.4 ± 0.8	ND
Kaempferol	27.56	60.6± 1.1	53.0 ± 0.1	89.3 ± 0.2	169.5 ± 0.7	334.4 ± 2.0	234.4 ± 1.0	111.1 ± 1.2	61.1 ± 0.1
Total		257.6	1963.7	1548.6	1201.6	1248.4	1110.6	859.7	910.5

Values are presented as means ± standard deviation (SD) of three replicates. ND: not detected.

**Table 3 molecules-27-03329-t003:** Identification of lipidic compounds in oils at increasing temperatures.

Compound	RT (min)	Mol. Formula	25 °C	40 °C	50 °C	60 °C	70 °C	80 °C	90 °C	100 °C
(e)-3(10)-caren-4-ol (%)	6.30	C_10_H_16_O	0.38 ± 0.02	ND	ND	ND	ND	ND	ND	ND
Cis-p-mentha-1(7),8-dien-2-ol (%)	7.82	C_10_H_16_O	0.03 ± 0.01	0.41 ± 0.02	0.02 ± 0.00	ND	ND	0.36 ± 0.02	0.39 ± 0.04	0.36 ± 0.07
Retinal (%)	11.20	C_20_H_28_O	4.43 ± 0.03	ND	0.04 ± 0.01	0.01 ± 0.00	0.02 ± 0.00	0.09 ± 0.01	0.01 ± 0.00	0.02 ± 0.00
Lycophyll (%)	22.77	C_40_H_56_O_2_	0.04 ± 0.00	0.05 ± 0.01	0.01 ± 0.00	0.01 ± 0.00	0.01 ± 0.00	0.02 ± 0.00	0.04 ± 0.00	0.04 ± 0.01
Androstatriene, 3-hydroxy-17-oxo (%)	46.26	C_19_H_24_O_2_	0.01 ± 0.00	0.03 ± 0.00	0.01 ± 0.00	0.02 ± 0.00	ND	ND	0.02 ± 0.00	0.02 ± 0.01
Carotene, 3,4-didehydro-1,2-dihydro-1-m Ethoxy (%)	14.54	C_41_H_58_O	0.02 ± 0.00	0.02 ± 0.00	0.2 ± 0.00	0.03 ± 0.00	0.03 ± 0.00	0.02 ± 0.00	0.04 ± 0.01	0.04 ± 0.001
Olean-12-en-3-ol, acetate (%)	49.82	C_32_H_52_O_2_	ND	ND	ND	1.55 ± 0.09	3.1 ± 0.23	4.32 ± 0.56	1.16 ± 0.11	1.18 ± 0.12
Amyrin (%)	49.78	C_30_H_50_O	ND	ND	0.15 ± 0.01	0.18 ± 0.03	0.2 ± 0.05	0.21 ± 0.03	0.08 ± 0.01	0.04 ± 0.00
2-Hydroxychalcone (%)	34.13	C_15_H_12_O_2_	ND	ND	0.03 ± 0.00	ND	0.03 ± 0.00	0.01 ± 0.00	0.02 ± 0.00	0.02 ± 0.00
Ethyl iso-allocholate (%)	21.88	C_26_H_44_O_5_	0.02 ± 0.00	0.03 ± 0.00	0.06 ± 0.00	0.02 ± 0.00	0.01 ± 0.00	0.05 ± 0.00	0.03 ± 0.00	0.02 ± 0.00
Spirost-8-en-11-one, 3-hydroxy (%)	51.78	C_27_H_40_O_4_	0.03 ± 0.00	0.01 ± 0.00	ND	0.01 ± 0.00	0.04 ± 0.00	0.03 ± 0.00	0.06 ± 0.01	0.03 ± 0.00
Betulin (%)	28.91	C_30_H_50_O_2_	ND	0.06 ± 0.01	0.05 ± 0.00	0.03 ± 0.00	0.01 ± 0.00	0.06 ± 0.01	0.06 ± 0.01	0.04 ± 0.00
Lupeol (%)	45.97	C_30_H_50_O	ND	ND	ND	0.01	0.01 ± 0.00	0.01 ± 0.00	0.01 ± 0.00	0.01 ± 0.00
Octamethyl-docosahydropicene-3,13-diol (%)	46.13	C_30_H_52_O_2_	ND	ND	ND	0.07 ± 0.01	0.01 ± 0.00	ND	0.02 ± 0.00	0.06 ± 0.01
Betulinaldehyde (%)	46.28	C_30_H_48_O_2_	ND	ND	ND	0.01 ± 0.00	0.03 ± 0.00	0.03 ± 0.00	0.08 ± 0.01	0.04 ± 0.00
Carbenoxolone (%)	46.32	C_34_H_50_O_7_	ND	ND	ND	ND	0.07 ± 0.01	ND	ND	0.07 ± 0.01
Astaxanthin (%)	14.05	C_31_H_50_O_3_	ND	ND	0.04 ± 0.00	ND	0.02 ± 0.00	0.05 ± 0.00	0.04 ± 0.00	0.01 ± 0.00
Urs-12-en-28-oic acid, 3-hydroxy-, methyl ester (%)	46.24	C_30_H_50_O_2_	ND	ND	ND	0.05 ± 0.01	0.03 ± 0.00	0.07 ± 0.01	0.06 ± 0.00	ND

## Data Availability

Data is contained within the article or Appendix A.

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
