# Peer review of "Energetic Bio-Activation of Some Organic Molecules and Their Antioxidant Activity in the Pulp of the Moroccan Argan Tree «Argania spinosa L.»"

_molecules, 2022, doi:10.3390/molecules27103329_

Round 1
Reviewer 1 Report
In this study, Energetic bio-activation of some organic molecules and their antioxidant activity of the Pulp of the Moroccan Argan Tree « Argania spinosa L » was carried out. This is an interesting work. However, some revision is required and the following points should be addressed:
- The abstract should be refined further.
- The HPLC chromatograms shoud be provided and the Methodological investigation data should be provided too.
- The GC-MS chromatograms should be provided.
- In 2.1. It is necessary to explain why heat treatment does not have a significant effect (p>0.05) on pH and titratable acidity,While the effect on insoluble solids is remarkable (p<0.05).
- In 2.2. Determination of color and chlorophylls, a temperature range should be indicated before the conclusion of the temperature increases the clarity of the powder.
- In line 153 of 2.3.1, the abbreviation of total flavonoids should be indicated in the text, which should be consistent with total phenolic compounds.
- In line 168, the change trend of TFC affected by temperature is not obvious. It is recommended to set the ordinate suitable for TFC on the right.
- It is suggested to replace the molecular weight in Table 3 with the molecular formula to reduce the interference to the following data.
Author Response
Dear Reviewer,
Our answers are provide in the attached document.

Reviewer 2 Report
The use of agri-food industry waste as a source of valuable biologically active ingredients is currently the subject of many studies. The aim of the presented study was to assess the effect of thermal treatments on the thermoactivation of argan pulp after extraction on its therapeutic value, including its therapeutic value.
antioxidant activity, its phenolic profile and functional and physicochemical properties. Nevertheless, some shortcomings were noted in the content. List of proposed changes below.
- For the sake of clarity of the text, I suggest inserting descriptions of the abbreviations used WAC, LGC, SC, WSI, OAC, TPC, TFC, PPO, CPTs, AAO, CPT, Ash after the keywords.
- In the text, the dry weight is marked with two abbreviations DM and DW proposes to stay with DM. Line 393 states: The results were expressed as mg per 100 g MS. Please indicate the correct value. Similarly, the text uses two abbreviations GAE, EAG meaning the same substance (gallic acid) proposes to stay with GAE.
- There is no need to repeat the significance level p <0.05 in the text, it is enough to mention it once in the chapter Statistical analysis.
- Color parameter L row 119 describes clarity rather than brightness level rather than clarity.
- Please provide the retention times for the analyzed compounds in tables 2 and 3.
- Chapter plant material description. 4.2.1 should be moved to chapter 4.1.
- In the "Statistical analysis " chapter, it is enough to provide the interval of null hypothesis testing at p <0.05, and in the text only using sentences, a statistically significant relationship was found or not found. In this chapter, the PCA test (Principal Component Analysis) should be added to the imaging of intergroup variability.
Author Response
Dear Reviewer,
Our answers are provided in the attached document.

Reviewer 3 Report
The manuscript presents the detail analyze of antioxidant activity, physicochemical properties, and chemical composition of a medicinal plant in the in the field of food and plant nutrition. The study presents comprehensive informations on the chemical content of Argania spinosa. The effect of heat treatment on the chemical composition and physicochemical properties were investigated in the study. The workflow and presentation of the manuscript looks good. The study is interesting and experimental data are meaningful.
However, I have some minor concerns about the manuscript according to the following reasons.
- Check the whole manuscript again. The language needs to be revised to enhance the quality of the work and the typographical errors should be corrected
- revise the full name of the plant as “Argania spinosa L.” in the several places (Argania spinosa must be italic, L. must be straight) .
- The units and standard deviation values must be given in all data?
- revise the Table 1. You used (,) in some results and (.) in some other, maket hem all (.)
- Most of the references are outdated publications, change some of them with new ones (later than 2020). Also, please use some other relevant following references in the introduction section in the phytochemical analysis and antioxidant activity of plants. "Chemical constituent and radical scavenging antioxidant activity of Anthemis kotschyana Boiss (2021). Natural Product Research 35 (22), 4794-4797" "Polyphenolic content, antioxidant potential and antimicrobial activity of Satureja boissieri. 2018. (IJCCE), 37(6), 209-219."
Author Response

(The authors gave the same response as above.)

Round 2
Reviewer 1 Report
The paper has been revised carefully and could be accepted in current form.